# The Future of Cardiothoracic Surgical Critical Care Medicine as a Medical Science: A Call to Action

**DOI:** 10.3390/medicina59010047

**Published:** 2022-12-27

**Authors:** Rafal Kopanczyk, Jesse Lester, Micah T. Long, Briana J. Kossbiel, Aaron S. Hess, Alan Rozycki, David R. Nunley, Alim Habib, Ashley Taylor, Hamdy Awad, Amar M. Bhatt

**Affiliations:** 1Department of Anesthesiology, Division of Critical Care, The Ohio State University Wexner Medical Center, Columbus, OH 43210, USA; 2Department of Anesthesiology, University of Wisconsin Hospitals & Clinics, Madison, WI 53792, USA; 3Department of Anesthesiology and Pathology & Laboratory Medicine, University of Wisconsin Hospitals & Clinics, Madison, WI 53792, USA; 4Department of Pharmacology, The Ohio State Wexner Medical Center, Columbus, OH 43210, USA; 5Department of Pulmonary, Critical Care & Sleep Medicine, The Ohio State University Wexner Medical Center, Columbus, OH 43210, USA; 6College of Medicine, The Ohio State University, Columbus, OH 43210, USA; 7Department of Anesthesiology, Division of Cardiothoracic and Vascular Anesthesia, The Ohio State University Wexner Medical Center, Columbus, OH 43210, USA

**Keywords:** cardiothoracic surgery, intensive care medicine, anesthesiology, delirium, acute kidney injury after cardiac surgery, ECMO, lung transplantation, transfusion medicine, ethics, ERAS, thoracoabdominal aortic aneurysm, spinal cord protection

## Abstract

Cardiothoracic surgical critical care medicine (CT-CCM) is a medical discipline centered on the perioperative care of diverse groups of patients. With an aging demographic and an increase in burden of chronic diseases the utilization of cardiothoracic surgical critical care units is likely to escalate in the coming decades. Given these projections, it is important to assess the state of cardiothoracic surgical intensive care, to develop goals and objectives for the future, and to identify knowledge gaps in need of scientific inquiry. This two-part review concentrates on CT-CCM as its own subspeciality of critical care and cardiothoracic surgery and provides aspirational goals for its practitioners and scientists. In part one, a list of guiding principles and a call-to-action agenda geared towards growth and promotion of CT-CCM are offered. In part two, an evaluation of selected scientific data is performed, identifying gaps in CT-CCM knowledge, and recommending direction to future scientific endeavors.

## 1. Introduction

Cardiothoracic surgical critical care medicine (CT-CCM) is a unique subspecialty that links diverse clusters of medical diseases to their perioperative management. As the burden of diseases requiring surgical intervention increases over the coming decades, CT-CCM expansion will follow in step. Cardiovascular diseases (CVDs) will be the predominant drivers of this growth. CVD prevalence nearly doubled between 1990 and 2019 [1]. With an aging population and improving socioeconomic status of underdeveloped countries, these trends are projected to continue [2]. By 2030, coronary heart disease and heart failure in the U.S. are forecasted to increase in prevalence from 2010 levels by 16.6% and 25%, respectively [2]. Consequently, procedures such as coronary artery bypass grafting (CABG) and therapies for end-stage heart failure such as durable mechanical ventricular device implantations or heart transplantations are expected to escalate [3]. These estimates hold true even when expansion of interventional cardiology is considered [4,5,6,7].

Other pathologic processes are also likely to affect the growth of CT-CCM. Chronic lung diseases such as chronic obstructive pulmonary disease (COPD) and idiopathic pulmonary fibrosis (IPF) are also on the rise [8,9]. Along with technological advances in lung preservation and extracorporeal support, this is likely to result in an increased number of intensive care admissions either after lung transplantation or extracorporeal membrane oxygenation (ECMO) cannulation for chronic respiratory failure [10,11]. Moreover, overall ECMO use is growing in adult populations, with the deployment of ECMO therapy for respiratory failure during the coronavirus disease 2019 (COVID-19) pandemic illustrating the unique expertise provided by CT-CCM intensivists [12]. With zoogenic spillovers of viruses becoming more likely with human encroachment on undeveloped wildlife habitats, the next global health emergency also has the potential to result in respiratory or cardiac sequela requiring mechanical organ support [13,14,15].

Consequently, an increase in patients requiring care in cardiothoracic surgical intensive care units (CT-ICUs) is likely, creating new stressors, placing pressure on current cardiothoracic surgical care systems. It will also force physician scientists to discover, innovate, and improve current therapies and treatments in order to address this growth. With these projections in mind, it is important to evaluate the current state of adult cardiothoracic surgical critical care science and to recognize missing data in the field in anticipation of the future rise in CT-ICU utilization. Initially, deficiencies, limitations, and barriers to advancement of the field need to be realized and addressed. Subsequently, knowledge gaps need to be appreciated and tackled. 

In this two-part review, we focus on concepts fundamental to the evolution of CT-CCM and patient care. First, we provide a list of guiding principles, aspirational goals, and a call-to-action agenda needed to promote and advance CT-CCM as a medical science. In part two, we assess current scientific data relevant to CT-CCM populations, identify selected knowledge gaps, and offer recommendations on investigations needed to bridge those gaps. 

## 2. Part 1—The Current State of Cardiothoracic Surgical Critical Care Medicine as a Medical Science

CT-CCM is a multidisciplinary endeavor composed of cardiothoracic surgeons, anesthesiologists, internists, pharmacists, perfusionists, and many more advanced level providers, nurses, and technicians (Figure 1) [2,16]. As such, the research pertaining to the CT-CCM patient populations is dispersed over many different professional societies and published in a variety of specialized journals. These include the American Thoracic Society, The Society of Thoracic Surgeons, Society of Critical Care Anesthesiologists, Society of Cardiovascular Anesthesiologists, American College of Chest Physicians, American Association for Thoracic Surgery, American Society for Artificial Internal Organs, Society of Critical Care Medicine (SCCM), and more, with each organization linked to an exclusive journal publication. Moreover, SCCM, heralded as a hub for all intensive care, seldomly embraces CT-CCM as a unique subset of critical care [17,18,19,20,21,22]. The resulting array of independent societies and the absence of a unified entity responsible for scientific oversight in the field creates compartmentalization of knowledge, hinderance to crosspollination of ideas, and deficiency in strategic planning. Additionally, it diminishes the visibility and advocacy for the field. Consequently, the current model is not conducive to coherent, systematic inquiry necessary for promotion and expansion of CT-CCM knowledge. Table 1 lists additional barriers to knowledge expansion in CT-CCM.

The status quo results in insufficient contribution of CT-ICU patients to the sample size of many landmark critical care trials, creating standards of practice that rely heavily on the application of data obtained from non-CT-ICU patient populations [23,24,25,26,27,28]. However, the pathophysiology and patient populations confronted in CT-CCM are unique and require exclusive scientific perspective. For example, determining the appropriate length of antibiotic therapy for a nosocomial pneumonia in patients with severe hypoperfusion related to postoperative heart failure or mechanical circulatory support (MCS) [29] cannot be answered by translating data from other ICU cohorts [30]. Pharmacokinetic features such as altered volume of distribution, or physiologic principles such as tissue penetrance with microcirculatory failure or continuous, versus pulsatile, blood flow, likely affect antibiotic length needed in CT-ICU populations [31,32,33,34,35,36,37]. These unique considerations need to be appreciated when developing standards of care and when designing clinical trials. Application of data from large, mixed population cohorts should be limited, and actively discouraged. However, these shortcomings create opportunities if a growth mindset is adopted; current knowledge gaps are a fertile ground for development of scientific investigations, improved clinical guidelines, and new career paths for medical professionals.

In order to cultivate CT-CCM as a unique science, the discipline requires establishment of foundational principles, as well as goals and objectives guiding future endeavors. Here, we propose major tenets and aspirational goals we find essential to the future of CT-CCM and its growth as a medical science, with major points summarized in Figure 2:

Major Tenets of Cardiothoracic Surgical Critical Care Medicine

Cardiothoracic Surgical Critical Care Medicine is a discrete subspeciality of a medical science.CT-ICU patient populations are diverse and medically unique.Distinct investigations enlisting CT-ICU cohorts are required to answer basic scientific or clinical questions relating to these populations.Data acquired from general medical or surgical ICU studies may not provide evidence easily translatable to CT-CCM. Application of such information should be done with caution.Wide knowledge gaps exist in many areas of CT-CCM.

Call-to-Action Agenda

Formation of a goal setting, centralized governing body, such as CT-CCM specific society.Establishment of a scientific journal centered on CT-CCM inquiry.Securement of funding and development of grant programs specifically geared towards CT-CCM research.Expansion of the Perioperative and Critical Care Conference co-sponsored by the Society of Thoracic Surgeons and the Society of Cardiovascular Anesthesiologists to include other stakeholders, such as Society of Critical Care Medicine, the American Association for Thoracic Surgery, the Society of Critical Care Anesthesiologists, and the American Academy of Cardiovascular Perfusion, and more.Establishment of a standardized CT-CCM training curriculum, continuing education, and certification.

It is our belief that the development of guiding principles and objectives is necessary for further growth of CT-CCM, and most importantly, for improvement of patient care. Such ambitious trendsetting is advised when innovation is required. Less effort may result in repetition of old patterns, stifling progress. Hence, further expansion of the specialty necessitates bold initiatives. Certainly, inaction is not an option.

## 3. Part 2 – Selected Gaps in Knowledge and Future Direction of Research

### 3.1. General Framework and Summary of Important Publications 

In part two of the review, we identify existing knowledge gaps affecting patients cared in cardiothoracic surgery intensive care unit (CT-ICU) and suggest a direction for further research in the field of CT-CCM. The list of topics is not exhaustive and intends to give sense of breadth and complexity of future work at hand. The discussion is divided into disorder-specific research considerations, followed by considerations for special populations. Figure 3 illustrates general framework of research needed to address CT-CCM knowledge gaps, and Table 2 lists and summarizes publications important to CT-CCM.

### 3.2. Disorder-Specific Considerations

#### 3.2.1. Cardiac Surgery-Associated Acute Kidney Injury (CSA-AKI)

Acute kidney injury is the most common major complication occurring after cardiac surgery with incidence reaching 40% [76]. It remains a significant cause of morbidity and mortality even 10 years after surgery and complete renal function recovery [76,77]. Although heavily researched, more questions than answers remain, including mechanism of injury, prevention, and management [76,77,78].

The pathophysiology of CSA-AKI remains incompletely understood with mechanisms such as ischemia, hypoxemia, reperfusion injury, hypoperfusion, inflammation, neurohumoral activation, extracorporeal circulation, genetic predisposition, and others causally identified [77,78]. These associations have not been verified in animal models, as logistical and cost factors have prevented development of cardiopulmonary bypass (CBP) animal studies [76]. This calls for further scientific pressure to improve the understanding of the processes and mechanisms behind CSA-AKI; an investment into development of a CSA-AKI animal model is prudent.

Many preventative and therapeutic measures have been deployed to combat CSA-AKI. Oxygen delivery, avoidance of hypoperfusion, and cardiac output augmentation continue to be the backbone of CSA-AKI prevention based the on current understanding of pathophysiology [76,79]. For example, goal-directed perfusion with oxygen delivery index >270–300 mL/min/m^2^ has recently been shown to improve CSA-AKI, but not in patients with high risk for postoperative AKI [79,80,81,82,83,84]. Other maneuvers have also been investigated. The Cochrane review on remote ischemic preconditioning in cardiac and major vascular surgery resulted in moderate to high certainty of no efficacy [85]. Additionally, most pharmacologic strategies have not proven to be helpful in preventing CSA-AKI. Therapeutics such as mannitol, steroids, dopamine, fenoldopam, sodium bicarbonate, theophylline, statins, N-acetylcystine, clonidine and antioxidant supplements have all failed to show renal-protective benefits [76,78]. Similarly, the type of resuscitative fluids used and their effects on kidney health have been debated. The majority of investigations for balanced versus normal saline crystalloid use come from mixed ICU populations, making their application to CT-ICU problematic [86,87]. Moreover, the use of albumin in continues to be controversial, with a mixed bag of results [88]. To date, the most significant positive results on prevention come from PrevAKI trials, where adherence to the Kidney Disease: Improving Global Outcomes (KDIGO) bundle reduced the rate of CSA-AKI in a single center RTC, and rate of moderate and severe AKI in a multicenter trial; the bundle consisted of avoidance of nephrotoxins, optimization of glycemic control, and optimization of volume status and hemodynamics [89,90].

Similarly, the ideal time of initiation of renal replacement therapy (RRT) has been in question [76]. Aside from well-established, life-threatening RRT indications, a single-center ELAIN trial showed benefit of early RRT initiation as compared to delayed initiation in a cohort of nearly 50% of patients recovering from cardiac surgery [76,91]. A retrospective, multicenter observational study in cardiac surgery patients showed similar results [76]. However, due to design flaws and lack of CT-CCM specific RTCs, the answer to early versus delayed RRT remains elusive.

In summary, many questions remain unanswered regarding CSA-AKI. An intense effort is needed to elucidate the mechanisms of renal injury associated with cardiovascular surgery, effective preventative measures, and well-timed and successful therapies.

#### 3.2.2. Delirium

Delirium is the most common neurologic complication encountered in the CT-ICU occurring in 7–50% of patients [92,93,94,95,96,97,98,99]. It is also a risk factor for long-term cognitive dysfunction, functional decline, and mortality, suggesting the continued necessity for research relating to this pathological state [93,94,97]. Further delineation of pathophysiology and mitigation of risk factors of delirium in CT-ICU patient populations are the two areas in need of urgent inquiry important for the development of preventative and therapeutic strategies.

Delirium is a syndrome with a common denominator of dysregulated neuronal function; however, systemic disturbances that produce this clinical entity are diverse [95,96,100]. Consequently, separate evaluation of pathophysiology of delirium in CT-CCM is required because of unique exposures such as profound hypothermia, non-pulsatile flow, or chronic hypoperfusion [96]. A tactic of creation of specific animal models is problematic due to difficulty of demonstrating the presence of delirium in experimental animals [95,96]. In addition, logistics and cost may be prohibitive. However, animal models with adequate face and construct validities would be helpful in identifying cellular and molecular changes [96]. Other innovative measures are also required to elucidate specific mechanisms behind delirium in CT-ICU populations. Further investigations into systems integration failure hypothesis, gut microbiome dysfunction, novel biomarker identification with biobank development, neuroimaging, and electrophysiology are warranted [100,101,102].

Additionally, a pragmatic research methodology is needed to address delirium as it relates to daily clinical practice. Tackling modifiable risk factors is most likely to yield clinically usable results. Focusing on patient-specific risks is especially important. Studies concentrating on optimizing preoperative frailty, physical conditioning, cognitive prehabilitation, nutritional status, hearing impairment, and chronic disease burden in CT-CCM populations are of utmost importance [93,95,99]. Cognitive prehabilitation is especially promising given encouraging results from non-cardiac surgery populations such as the Neurobics trial [103]. A recent feasibility trial of perioperative cognitive training in cardiac surgery established it as a viable target for further investigations [104]. Likewise, chronic disease burdens such as depression, arrhythmias, diabetes mellitus (DM), hypertension (HTN), stroke history, peripheral vascular disease, and obesity have been found to be statistically significant risk factors for post-CABG delirium and additional studies are needed to evaluate optimization of chronic diseases as a preventative measure [99].

Moreover, investigations of modifiable precipitating factors such as postoperative pain control and sedation are crucial. Both uncontrolled pain and excessive opioid administration are significantly related to delirium development [92,93,95,96]. Recent years flourished with new regional anesthetic techniques involving fascial spread of local anesthetics, allowing anesthesiologists to develop blocks covering previously inaccessible dermatomes [105]. Aggressive evaluations of erector spinae, serratus anterior, and transversus thoracis muscle plane blocks with randomized controlled trials (RTCs) in CT-CCM populations are needed to determine their effectiveness for optimal chest wall pain control [106].

Sedation choice is another modifiable risk factor needing further examination. Initial positive reports of dexmedetomidine in cardiac surgery have recently been eclipsed by strong RTCs questioning its benefit [94,98,107,108]. As a result, there continues to be no single sedative agent deemed helpful in delirium treatment or prevention in any realm of critical care. An application of volatile anesthetics for ICU sedation is on the horizon. Inhaled agents have been used for sedation with success in Europe for some time, and now, isoflurane will be evaluated in the U.S. in phase 3 clinical trials INSPiRE-ICU1&2 (NCT05327296) [109,110]. Theorized benefits include decreased opioid use, improved spontaneous breathing, shorter extubation times, and quicker wakeups [111]. INSPiRE-ICU 2 will also evaluate long-term cognitive outcomes. Pending results, CT-CCM cohorts will be another frontier for inhaled anesthetic inquiry. Finally, an ongoing study of lidocaine infusion for COVID-19 ARDS with secondary endpoints of delirium and opioid consumption may further aid clinicians in sedation selection and pain control (NCT04609865) [112].

In summary, investigational strategy into delirium in CT-CCM patient populations should follow a two-pronged response. Basic science research should address mechanisms of delirium in specific clinical scenarios, while pragmatic clinical studies should be designed to help mitigate risk factors of delirium.

#### 3.2.3. Pharmacotherapy

Gaps in knowledge exist in CT-CCM in relation to disease-specific pharmacotherapy. With the appreciation of the magnitude of the missing data, we highlight just a few pieces of a puzzle in the following section.

Patients undergoing CABG are likely the most common patient population seen in the CT-ICU. Frequently, grafts used include saphenous-vein grafts and radial artery grafts. The RADIAL trial compared saphenous vein grafts to radial artery grafts for CABG and demonstrated a significantly lower rate of adverse cardiac events in patients who received radial grafts (hazard ratio 0.67; 95% CI. 0.49–0.90; *p* = 0.01) [113]. However, the radial artery is more muscular in nature, increasing concern regarding vasospasm leading to myocardial ischemia [114]. As a result, the RADIAL trial had six different regimens to manage to prevent arterial graft spasm that differed in agents (diltiazem, nifedipine, or amlodipine) and duration (6 weeks to indefinite) [113]. Consequently, the quest for antispasmodic medications to help prevent vasospasm is ongoing. Medications that have been studied include nitroglycerin, diltiazem, verapamil, papaverine, and milrinone [115]. However, many of the investigations are limited to single centers with small sample sizes. Due to the lack of conclusive evidence, the ideal agent and duration of its use remain in question [116].

Another area with a paucity of literature is optimal antibiotic management for delayed sternal closure (DSC). Although guidelines give recommendations on agent, dose, and duration of antibiotics for prevention of surgical site infections (SSI), there are no recommendations on antibiotic therapy if primary closure is not performed [117,118]. Feared complications of DSC are the deep sternal wound infection and mediastinitis. Thus, antibiotic prophylaxis is commonly continued, but the regimens and durations vary depending on institution [119,120]. Two recent trials showed that continued administration of prophylactic antibiotics in patients with DSC were not associated with benefits in rates of mediastinitis and deep tissue infections [120,121]. Both studies were limited by their retrospective nature, single center location, and small sample size. Additional studies are needed to determine if prolonged antimicrobial prophylaxis is needed and if so, what the optimal regimen in patients managed with DSC is.

An additional area that presents frequent clinical conundrums in the CT-ICU involves appropriate dosing of pharmacotherapy for patients on extracorporeal membrane oxygenation (ECMO). The available literature has revealed that patients on ECMO have an increased volume of distribution and elimination of certain medications [122]. Drugs with high lipophilicity or protein binding have demonstrated decreased blood concentrations, likely due to drug sequestration in the ECMO circuit, putting the patient at risk for clinical failure [123,124]. Most of the literature investigating optimal dosing are limited to ex vivo data or case reports and case series [122]. Due to the limitations of the literature, the provider is presented with the task of weighing the risk of an adverse drug event and the risk of clinical failure. Where this may be of most concern is the dosing of antibiotics for patients on ECMO as clinical failure could be detrimental. A recent publication comparing serum concentrations of various antibiotics in patients on ECMO versus medical therapy demonstrated a higher rate of failure to reach target concentrations of piperacillin (48.4% vs. 13.0%) and linezolid (34.8% vs. 15.0%) [125]. Previously, the medications that were lipophilic or highly protein-bound raised concern for therapeutic failure. However, neither piperacillin nor linezolid are particularly lipophilic with relatively low protein binding, making it less likely that sequestration in the ECMO circuit is responsible for failure to attain target concentrations. The Surviving Sepsis Campaign [126] recommends therapeutic drug monitoring (TDM) in critically ill patients due to alterations in volume of distribution and elimination. For patients on ECMO, TDM becomes imperative as the addition of ECMO adds another complexity to dosing considerations in an already critically ill patient. However, many institutions do not have the ability to provide TDM for many of these medications. Future studies should focus on the use of TDM for patients on ECMO and the effect of alternative dosing regimens in obtaining therapeutic goals.

In conclusion, significant knowledge gaps exist in relation to pharmactotherapy specific to CT-ICU patient populations. Provided examples serve as a sample of a work ahead. However, other important pharmaceutical concepts deserve to be accounted for in this review. These include:-Therapy for vasoplegia after CBP;-Vasopressor of choice for hypotension;-Inotrope of choice based on pathology;-Utility of a calcium sensitizer;-Antibiotic therapy duration for hospital-acquired infections in cardiogenic shock;-Pathology and mechanical support specific anticoagulation regimens and reversal agents;-Nalaxone and spinal cord protection;-Multimodal analgesics.

#### 3.2.4. Transfusion and Blood Conservation

One in five cardiothoracic surgery patients will receive blood [127]. Unlike many fields, optimal red cell transfusion thresholds are relatively well-defined in CT surgery. For the general cardiac surgical population, a threshold of <7.5 g/dL is safe: a 5243-participant study of a <7.5 g/dL versus a <9.5 g/dL trigger for transfusion during and after moderate-to-high-risk cardiothoracic surgery found reduced transfusions and no evidence of harm in the restrictive group [128], with consistent outcomes at six months [129]. It is quite likely that a threshold of 7.0 is comparable to 7.5, but that has not been explicitly studied in large cardiac surgery trials. For non-surgical patients with active ischemia, a threshold between 7–8 g/dL is likely safe but not firmly established: a trial of 668 patients with anemia and myocardial infarction found that a transfusion trigger of <7 g/dL was noninferior to a trigger of 10 g/dL for major adverse cardiac events at 30 days—although the confidence interval for this result may include clinically significant harm [130]. The results of the 3500-patient Myocardial Ischemia and Transfusion (MINT) trial, expected to conclude in 2023, will likely clarify the safety of restrictive transfusions in patients with active myocardial ischemia [131]. Other blood products are not well-studied in the CT-ICU or any field. Triggers for plasma, cryoprecipitate, and platelet transfusion suggested by various cardiothoracic societies are reasonable but largely based on expert opinion or low-quality evidence [132]. Limited new data imply that the traditional emphasis on early platelet transfusion to overcome CPB-associated platelet dysfunction and consumption is misplaced and that fibrinogen repletion is more clinically effective [133,134]. Cold-stored human platelets may offer better efficacy and safety than traditional room-temperature-stored platelets, and the results from an ongoing trial in complex cardiac surgery are eagerly anticipated [135]. Alternatives to transfusion have also been studied: multiple meta-analyses of acute normovolemic hemodilution find that it reduces red blood cell transfusion by around 0.75 units per case and mildly reduces blood loss [136]. However, published trials are highly heterogeneous and the practice remains controversial [137]. Finally, factor concentrates may be reasonable alternatives to transfusion in cardiac surgery: a trial of 827 patients found that using fibrinogen concentrate was non-inferior to cryoprecipitate for post-CPB bleeding associated with hypofibrinogenemia [138], and similar studies are planned to compare 4-factor prothrombin complex concentrates with plasma [139].

Laboratory testing is a critical adjunct to transfusion practice in the CT-ICU. Guidelines from the Society of Cardiovascular Anesthesiologists and other organizations have placed significant emphasis of the benefits of laboratory-guided transfusion algorithms [132], which reduce transfusions and bleeding compared to physician judgement [140]. The relative benefits of specific coagulation testing platforms remain unproven and hotly debated—trials purporting to show the superiority of viscoelastic systems such as thromboelastography (TEG) over conventional coagulation tests have been confounded by the manner in which the tests were performed for the trial, e.g., comparing beside TEG to coagulation tests performed in a distant laboratory [141]. Well-run trials in which logistical confounders were eliminated have failed to find a relative benefit of one testing platform over another [142]. This suggests that testing algorithms should be designed to emphasize whatever platform is most pragmatic at the specific institution, without undue preference for a specific method. One exception to this may be in the case of a patient who has taken antiplatelet agents, where modified viscoelastic assays such as TEG Platelet Mapping may provide some useful information to guide timing and dosage of transfused platelets [143]. In the near future, rapid advances in point-of-care genomic and epigenetic testing may also be used to identify patients with differential responses to antiplatelet and anticoagulant medications, as well as responsiveness to transfusion therapy [144,145].

#### 3.2.5. Paralysis after Aortic Aneurysm Surgery

Patients who receive surgical or endovascular treatment for thoracic aortic aneurysms and/or dissection are often in the CT-ICU for several days to weeks. A contributing factor for longer ICU stays in these patients are post-operative complications. One of the most devastating complications of either open surgical or endovascular repair of aortic aneurysms is spinal cord ischemia and/or paralysis. Although both treatment paradigms have a risk of postoperative paralysis, there are different mechanisms which govern the pathophysiology of open aortic surgery and thoracic endovascular aortic repair (TEVAR) mediated paralysis [146]. A recently published article demonstrated in a large animal model that these two mechanisms are different. Namely, the open surgical approach causes ischemic reperfusion injury versus critical permanent hypoperfusion in endovascular repair. A preventative therapeutic measure to treat these different phenomena will require an animal model that accurately maps the different pathophysiology of ischemic spinal cord injury in open and endovascular repair in humans. However, this new information needs to be confirmed in humans and other animal models.

In our small and large animal models, transient aortic clamping during open surgical repair has been shown to cause primarily grey matter damage via reactive oxygen species mediated blood–spinal cord barrier disruption and leakage of intracellular contents, causing central cord edema and neuronal death. TEVAR, on the other hand, has been shown to cause white matter damage likely due to chronic hypoperfusion of segmental arteries by the stent graft. CT-ICU management of these patients currently consists of spinal cord drains to reduce spinal cord edema and the usage of vasopressors to maintain systemic perfusion pressure. The treatment of paraplegia after spinal cord ischemia requires further mechanistic clarity and basic science research to develop a pharmacologic treatment to either prevent ischemic spinal cord injury perioperatively or reverse it postoperatively. Additionally, these treatments should be tailored to the specific patient, keeping in mind the procedure they underwent and the specific mechanism of spinal cord damage.

Which animal model best replicates the pathophysiology of paraplegia after open and endovascular repair? It is well known that the blood supply to the spinal cord varies across species, which raises the question of which model is the most appropriate to conduct basic science research [147]. Once the ideal animal model is developed to test therapeutics, there is another obstacle which researchers must consider, the mismatch which often occurs between the size of the lesion within the spinal cord observed on imaging and the functional symptoms of the lesion in the spinal cord. Simply put, lesions of similar size can result in different clinical outcomes. Termed the “neuroanatomical functional paradox”, this phenomenon is a persistent barrier to designing animal models of spinal cord injury [148]. This paradox makes it especially difficult to develop therapeutics, because certain regions of the spinal cord have higher levels of “eloquence” and have higher sensitivity to detect the effectiveness of treatment.

On the clinical side, currently the standard of care is to drain the cerebrospinal fluid (CSF) and to manage the systemic perfusion pressure, despite low evidence and small studies demonstrating the efficacy. Even the definition of high and low-risk is not well delineated. This is a hurdle for clinicians who aim to design randomized controlled trials, given the perceived benefit of the spinal drain. Additionally, the complication associated with the spinal drain is significant, requiring clinicians to weigh the risk of prophylactic spinal drains with its benefits. A solution to this problem is a randomized controlled trial which analyzes the efficacy of prophylactic spinal drains versus their complications.

Finally, there is a need for biorepositories using biological fluids (e.g., blood, CSF, urine) of patients who develop paralysis after open and endovascular aortic repair. The goal of such an endeavor will be to gain mechanistic clarity and allow researchers from many institutions to contribute to and acquire data from this repository. This can potentially fuel drug discovery targeting specific mechanisms which are involved in the pathogenesis of spinal cord injury and/or paraplegia after open and endovascular aortic repair.

There is a gap in the knowledge in the field of spinal cord perfusion and that addressing the following four points is of paramount importance:(1)What is the ideal small or large animal model for open and endovascular repair of aortic aneurysms, given the variety of anatomical blood supply to the spinal cord across species?(2)Given the neuro-radiological-anatomical functional paradox, therapeutic treatment for both disease paradigms is different and there is a need for more preclinical trials targeting the specific mechanism behind the grey- and white-matter lesions.(3)There is a need for randomized controlled trials testing the efficacy of spinal cord drains perioperatively to prevent paralysis.(4)There is a need for a repository containing the biological fluids of non-paralyzed patients as well as patients who develop paralysis after aortic interventions to gain mechanistic insight which will guide pharmaceutical discovery in this field.

#### 3.2.6. Cardiac Surgical Unit—Advanced Life Support

Cardiac Surgical Unit—Advanced Life Support (CSU-ALS) was conceived as a more focused resuscitation protocol for the specific needs of post-sternotomy patients, as compared with standard ACLS. The impetus for the protocol was the recognition that external cardiac massage is best avoided or minimized in post-sternotomy patients as they are more exposed to injury from this method of artificial perfusion. In addition, the most common causes of cardiac arrest after cardiac surgery (malignant arrhythmias, bradycardia, cardiac tamponade, bleeding) are immediately reversible with appropriate recognition and treatment [149]. The goal of rapid re-sternotomy if initial resuscitation attempts fail is another hallmark of the protocol. The rationale for this is multifaceted. First, several common causes of arrest are either alleviated with re-sternotomy (cardiac tamponade) or more readily treatable (bleeding source recognition, epicardial pacing wire dislodgement, arrhythmias treatable by internal cardioversion). Second, the recognition of the superiority of internal cardiac massage to external massage with respect to perfusion pressures and rate of ROSC is another benefit of rapid re-sternotomy. Finally, the right ventricular injury from the posterior sternum during chest compressions has been described, calling for development of less injurious maneuvers [150,151].

The CSU-ALS protocol gained significant validation when a panel of experts from the Society for Thoracic Surgeons (STS) endorsed it in 2017 [149]. This prompted a wide-spread desire to implement CSU-ALS in cardiothoracic surgical ICUs world-wide. As a novel approach to resuscitation in this unique patient population, this presents significant opportunity for research and improvement on every aspect of the protocol. Given the prevalence of cardiac disease and need for surgical intervention via sternotomy, any improvements in this protocol have the potential to result in dramatic improvements in outcomes for a large patient population. In addition, with its relatively recent endorsement by the STS, there is a significant need for wider awareness and training in hospitals in the practice of the CSU-ALS protocol. Finally, strong scientific effort is necessary to further study CSU-ALS approach. Inviting American Heart Association to join future endeavors would help to achieve both, wider awareness, and research development.

## 4. Special Populations

### 4.1. Extracorporeal Membrane Oxygenation for Respiratory Failure

ECMO utilization for respiratory failure refractory to medical management has expanded [152,153,154], yet many questions remain regarding potential benefits, including appropriate implementation and management [155]. First, limited randomized work exists elucidating selection of patients with respiratory failure that may benefit from ECMO. Patient selection continues to focus on disease severity assessed by hypoxia, respiratory acidosis and mortality risk [156,157]. However, risks and benefits of both ECMO and conservative management are patient-specific [156,157]. Comparison of survival scores for ARDS with the use of ECMO (the RESP score [158]), and without ECMO (the Murray score [159]) can be insightful, yet additional considerations should include relative contraindications [156] and consideration of pre-morbid and active clinical characteristics. These include age, body mass index, disease comorbidities, right ventricular function, [160] ventilatory compliance, [161] clinical timing and more. Additional research is required to offer highly patient-specific guidance towards appropriate implementation of ECMO.

The appropriate timing of ECMO initiation also remains unclear and may impact outcomes. Early initiation may improve oxygen delivery while minimizing ventilator-induced lung injury (VILI). Later initiation, however, may offer time to respond to conventional therapy and avoidance of ECMO-specific risks. “Early” initiation of ECMO starts within 3–6 h of initiation of intensive therapy [156,157], and is advocated for by the Extracorporeal Life Support Organization (ELSO). Presently, this is supported by limited randomized [157,162] and observational work [163]. Nonetheless, well-designed and powered trials comparing modern-conservative management (traditional measures with later “rescue” ECMO) to early ECMO institution are required [162].

Conversely, the tail-end of acceptable ECMO initiation is generally considered 7–10 days, noting that VILI progression and/or other end-organ injury may limit potential benefits. Nonetheless, reports do suggest that later initiation may be acceptable as a rescue strategy [164]. A strict cutoff, therefore, is ill-defined and trials have poorly dictated the benefits, or lack thereof, of “late” ECMO initiation. More guidance is urgently needed, highlighted by the COVID-19 pandemic, where initial hypoxia with reasonable ventilatory compliance is common, but late deterioration and worsened ventilatory parameters may have prompted late consideration of ECMO [165].

Moreover, knowledge gaps in the appropriate management of ECMO exist. First, the ideal ventilatory strategy while on ECMO is yet to be defined. Certainly, adamant avoidance of VILI is critical in order to maintain benefits of ECMO utilization [166]. Evidence is needed to dictate optimal PEEP and F_I_O_2_ strategies, ventilatory approaches to minimizes self-induced lung injury and consideration for prone positioning [166,167]. Next, research is needed to define the optimal anticoagulant drug, dose, and monitoring strategy to prevent bleeding and thrombotic events for patients on ECMO [168,169,170,171]. Some patients may even benefit from the exclusion of anticoagulation altogether [172]. Next, selecting an optimal cannulation approach, including dual versus single cannulation strategies with or without right ventricle support, remain challenging [173,174]. These are further emphasized since some approaches could enable unique “wearable” membrane oxygenators, one potential future direction of ECMO support [175]. Other approaches, such as extracorporeal carbon dioxide removal (ECOR), offer some unique benefits in the face of significant weakness and remain without clear guidelines or consensus [176]. Even integration of dialysis circuits into ECMO remains actively debated [177].

Lastly, other practice guidelines and general critical care considerations must be applied to patients on ECMO. This includes strategies towards reducing nosocomial infections in the ICU, including unique considerations for patients on ECMO [178], and the ethics of end-of-life care while on significant organ support [179].

### 4.2. Extracorporeal Cardiopulmonary Resuscitation (E-CPR)

Another area that warrants expertise and experience to provide rapid deployment of an immense resource pool is E-CPR. Compared to conventional CPR (C-CPR), E-CPR can markedly improve cardiac output to normal or near-normal levels. E-CPR can therefore serve as an expedient and effective bridge to workup, intervention and/or decision that, compared to conventional CPR, may improve mortality and neurologic outcomes by limiting low-flow states related to C-CPR compression quality and ischemia-reperfusion injury (IRI) related to C-CPR pauses in compressions [180,181]. Furthermore, E-CPR, once initiated, can maintain blood flow without compression-induced chest and cardiothoracic trauma.

Importantly, patient benefits have yet to be fully elucidated; studies remain underpowered and have not shown convincing, statistically significant improvements in survival and neurologic outcomes [182]. This is critical as E-CPR requires extensive, rapid resource deployment with heavy infrastructure and cost considerations [183,184]. Furthermore, appropriate implementation considerations exist, including the approach to transport during C-CPR, appropriate timing of initiation after C-CPR begins [182,185], and initial anticoagulation strategies. The latter issue is particularly nuanced as approximately 50% of patients suffering from arrest where E-CPR is utilized meet disseminated intravascular coagulation criteria at the time of cannulation [186] yet many require intense anticoagulation to support percutaneous coronary intervention.

### 4.3. Enhanced Recovery after Surgery (ERAS)

ERAS protocols are multidisciplinary and multimodal initiatives designed to shorten a patient’s entire perioperative journey through reduction in complications and a faster return to normal activity [187,188]. Various surgical subspecialties have validated evidenced-based ERAS protocols. However, such consistent and reproducible protocols for cardiac surgery are lacking in the literature. There are a multitude of challenges and limitations in developing such protocols, with the most frequently cited being an understanding that cardiac surgery patients have more complex pathologies and comorbidities compared to other sub-specialty surgery populations and undergo more diverse and invasive surgical approaches. This recognized complexity led to the recent formation of the ERAS Cardiac Society, whose guidelines and manuscript summarize key elements from prior studies marked with the class of evidence and level of recommendation to advise future practice [189].

Despite the promising results of such emerging guidance, ERAS protocols are not only a matter of application but require a willingness to change and break with long-established patterns. Some of the new patterns identified within cardiac surgery involve the changing patient population as already previously identified. Using coronary artery bypass grafting (CABG) as an example, the mean age of patients undergoing such surgery has increased from 58.3 to 68.5 years, with 38 percent of patients being 70 years or older with an average ejection fraction of 35 percent [190]. Since the beginning of the century, observational, risk-adjusted, and propensity-matched studies have further documented an increased mortality after CABG in women compared with men for all age-adjusted groups [191]. Future ERAS protocols for cardiac surgery should therefore consider the specific needs of the growing elderly population and women as they traverse their perioperative journey to decrease their proven incurred risk of heightened morbidity and mortality.

The goal of cardiac surgery in elderly patients has been described as focused on the improvement of quality of life rather than in prolonging life expectancy [192]. Influencing variables for such a focus have been shown to be related to a patient’s individual pre-operative nutritional and functional status. Three specific areas of interest in ERAS protocols with only minimal to moderate quality evidence and research include preoperative measurement of albumin for risk stratification, preoperative correction of nutritional deficiency, and prehabilitation. Hypoalbuminemia, for which the elderly are already at risk, has been shown to be a prognostic indicator of increased time on a ventilator, length of hospital stay, infection, and recovery from acute kidney injury [187,193,194,195]. Additionally, there are currently no high-powered trials looking at the benefits of early nutritional therapy for elderly cardiac patients who are considered high risk, while ERAS protocols for colorectal surgery have shown benefits of pre-supplementation to include a reduction in prevalence of infection post-operatively [196]. Low albumin and other nutritional markers are associated with greater morbidity and mortality in cardiac surgery patients, but there is minimal quality support for their strong recommendation and application in ERAS protocols, with further knowledge gaps existing regarding their specific importance in the elderly population. Regarding prehabilitation, three non-cardiac surgery studies have demonstrated benefits in functional capacity and reduction in postoperative complications in the context of ERAS [197,198,199]. Still, such interventions prior to cardiac surgery have yet to be examined to enable advancement of this area of ERAS research as well.

An additional unique circumstance that can influence recovery after surgery in the elderly population involves caregivers. Caregivers of elderly patients have an expertise that stems from their lived experience with the daily impact of cardiac disease and resulting therapeutic burdens [200]. This places them with a strategic position to provide insights into the development of clinical practice guidelines. Therefore, it would be beneficial to include such perspectives in the development of ERAS protocols for elderly patients after cardiac surgery in the future.

Another vulnerable population undergoing cardiac surgery is women. Cardiovascular disease is the leading cause of morbidity and mortality for women in the United States and worldwide [201]. Differences in surgical outcomes between men and women are multifactorial. Frequent explanations in the literature as to why women suffer greater cardiac morbidity include that they often present more acutely, have a delayed diagnosis, or angiographically lack significant coronary artery disease early in their disease process [202]. Even despite this lower predisposition to develop visible atherosclerosis and aortopathy, women have been shown to be at least three times more likely to rupture or dissect a thoracic aneurysm [203]. A multicenter retrospective study showed that women had worse outcomes for both elective and emergent cardiac valve surgery with prolonged length of CT-ICU stay and higher rates of respiratory failure [204]. Additionally, in the well-known EXCEL trial, women who were older with more comorbidities at the time of revascularization had increased procedural and post-operative complications [205]. The increased risk for women undergoing cardiac surgery necessitates their specific inclusion and acknowledgement of specific risk in ERAS protocols. There has also been significant attention brought to referral, diagnostic, and research bias within the medical community with disastrous implications that are not limited to cardiac surgery [191]. In a multicenter review by the American College of Cardiology, there was continued underrepresentation of women in cardiovascular research ranging from basic science studies to difficulties in enrolling women in cardiac surgery trials [191]. Due to women’s increased risk and lack of adequate management data, sex-based ERAS protocols and algorithms should also be part of further research in cardiac surgery.

### 4.4. Lung Transplantation

Since the performance of the first lung transplant procedure over fifty years ago, continued advancements in the field have resulted in not only more transplants being performed but also in expanded opportunities for patients who may not have previously qualified. Indeed, even older patients with several comorbidities who would not have been candidates a decade ago are now receiving and benefiting from lung transplantation [206,207,208]. However, as their complexity has increased, there has been a comparable need to be able to manage these comorbidities, which in turn has required an increasing investment from critical care services. Not only has this challenged the intensivist, but also respiratory therapists, nutritionists, physical therapists, and others who contribute to the multi-disciplinary structure of the modern intensive care unit.

In an earlier era of lung transplantation, patients with advanced lung disease already committed to mechanical ventilation were universally deemed not to be candidates for transplant. Following some successes with offering transplant to younger patients, dependency on mechanical ventilation is no longer considered an absolute contraindication for the procedure [209,210,211]. Indeed, even more aggressive respiratory interventions such as ECMO can now be employed in support of potential lung transplant candidates. The latter circumstance has been exemplified during the recent SARS- CoV-2 pandemic when many patients requiring emergency ‘rescue’ lung transplant needed temporary ECMO support as a ‘bridge’ to that procedure [212,213,214]. Additionally, the initiation of ECMO using a dual lumen cannula placed via the internal jugular vein has allowed for selected candidates to receive needed physical therapy and ambulate prior to transplantation [215,216]. In addition to pulmonary support via ECMO, other end-organ support such as invasive RRT and ventricular assist devices may be utilized in candidates where dual organ transplantation is being considered (i.e., heart–lung or lung–kidney) [217,218,219,220]. As these therapies must be employed in the ICU setting, the modern intensivist must be facile with their use.

Once successful lung transplantation has occurred, recipients return to the ICU for immediate post-operative management where the challenges to the intensive care team continue. These include complex ventilator management, hemodynamic support and fluid resuscitation, the initiation of systemic immunosuppression with its potential to endow significant infections, and possibly the need for continued or additional mechanical organ support (i.e., cardiac and renal) [221,222,223]. Among these, appropriate ventilator management is particularly important as the goal is to liberate the recipient from this support as soon as is feasible. Heightened expertise in this regard is typically necessary as the lung allografts of new transplant recipients can be complicated by ischemia reperfusion injury (i.e., primary graft dysfunction), pleural fluid collections (including hemothorax), diaphragm dysfunction, stenosis of the vascular anastomoses, and complications of the bronchial surgical anastomoses [224,225,226,227,228,229]. Additionally, continued ECMO support may be necessary until the lung allograft(s) mature following the implantation surgery.

In some instances, recipients may have a prolonged need for post-surgical ventilator support, and in these situations the expertise of extended members of the ICU team becomes especially important. As many of these recipients may have entered the transplant surgery with some decrement in overall physical conditioning and reduced muscle mass, the ICU nutritionist must be particularly attentive to their early post-transplant nutritional needs. This is necessary to prevent further compromise of respiratory muscle mass and resulting dysfunction. To further assist recovering muscle function, appropriate physical therapy must also be initiated in the ICU setting, often with the recipient still requiring mechanical ventilation [230,231]. Once liberated from the ventilator, recipients must be assessed for any speech and swallowing difficulties; the latter being particularly important as aspiration of swallowed matter (food or medications) can adversely affect the new lung allograft(s).

Lastly, in addition to the critical care management of lung transplant candidates and recipients, the intensivist has recently assumed an increasing role in the management of potential organ donors. Indeed, studies have suggested that the continued involvement of an intensivist and delivery of critical care services following declaration of neurologic death typically results in an overall increase in the procurement of not only lungs suitable for transplantation, but other organs as well [232]. For the intensivist, this requires a change in the overall approach to the potential deceased organ donor by shifting treatment away from one that is designed to support cerebral perfusion to one that prioritizes individual organs. One example of this would be a move away from the use of hyperventilation and osmotic diuretics for the treatment of cerebral edema in favor of a more liberal fluid resuscitation strategy aimed at supporting renal and hepatic function. Furthermore, as there is a significant endocrinopathy associated with the onset of neurologic death, hormonal resuscitation of the potential donor (as can be provided by the intensivist) has been shown to also enhance the suitability of all organs for transplantation [233,234]. Recently, the Society of Critical Care Medicine has published guidelines for the management of potential organ donors and has encouraged continued involvement of intensivists in this endeavor [235].

## 5. Ethical Considerations

The critical care patients encountered in the CT-ICUs present a unique challenge that pushes the bounds of medical ethics in modern hospital systems. These specific populations are often admitted following surgical procedures and possess some of the highest morbidity and mortality encountered in medicine. This often creates ethical conundrums. First, patients frequently undergo these procedures without prior planning for the possibility of being unable to make future decisions on their own. Second, families compound undue burden and stress as they are often unaware of patient’s wishes [236]. Moreover, medical teams, frequently inexperienced in appropriate management of difficult social or end-of-life situations, promote physician-driven decisions [237,238]. Furthermore, publicly reported quality metrics such as 30-day mortality after general cardiothoracic surgery and 1-year mortality after transplantation can be at odds with patient/family goals [239,240,241]. Finally, timing of therapy cessation, defining “futility”, and withdrawal of care while on extensive mechanical support can also be encountered. These problems often result in significant ethical dilemmas, especially when continuation of therapies might not have been chosen by the patient in the first place.

Patient autonomy is of paramount importance; therefore, patients or their families have the right to refuse therapies at any point. This can be incredibly painful to the team who has invested time, effort, and care into the case. However, the concept of reciprocal obligation of the patient to continue treatment until deemed futile by the medical team after the health system has invested significant resources for their care is paternalistic. Ultimately, in a perfect world without limits on the availability of medical care, we would fully support patients to the extent that they desire, everyone would always be capable of expressing their wishes, and all patients would have an educated family with excellent coping mechanisms to handle end of life decisions making, with all practitioners endlessly skilled in communicating with every family. Unfortunately, this world does not exist, and medical teams should act in accordance with patients’ wishes, maintaining their autonomy.

When working through philosophical and ethical problems, a consensus decision between the patient, support persons, and care team is the best and the most practical answer. One solution is for medical teams to add conflict management to their clinical skills during medical training. Active commitment to learning communication and conflict resolution, along with an ethical lens for our clinical care, can benefit practitioners, patients, and medical systems [242,243]. Finally, increasing the utilization of palliative medicine specialists can aid both families and medical teams significantly [244,245].

In summary, further scientific work is needed to guide education of future physicians in communication skills, conflict resolution, and clinical application of ethical principles. Additionally, clinically active care teams should investigate how investment of time and effort to train their members in these areas benefits their patients with quality improvement projects. Finally, we recommend a development of close relationships with palliative medicine colleagues and encourage their frequent involvement. Research is needed to further elucidate the value of palliative team involvement with all patients coming through CT-ICUs.

## 6. Conclusions

The demand for cardiothoracic surgical critical care will continue to expand as the burden of chronic diseases increases and the population ages. In result, utilization of CT-ICUs will escalate. This calls for reevaluation of care provided in these ICUs in preparation for the surge. Given the diversity of disciplines practicing CT-CCM, expertise specific to the field is dispersed over many different specialties, societies, and journals. Resultant compartmentalization of scientific inquiry without centralized governing body hinders growth, innovation, and patient care. Significant knowledge gaps exist in CT-CCM with significant portion of current CT-ICU care standards relying on data obtained from medical or general surgical patient cohorts. To improve the current model, CT-CCM needs to be recognized as a its own subspecialty and science. Additionally, subspeciality specific society, a journal publication, comprehensive annual meeting of all the stakeholders, and unified research agenda are necessary to promote its expansion. The future of cardiothoracic surgical critical care depends on realization of its distinction, unification of its professionals, realization of its knowledge gaps, and initiation of research required to answer the most pressing questions.

## Figures and Tables

**Figure 1 medicina-59-00047-f001:**
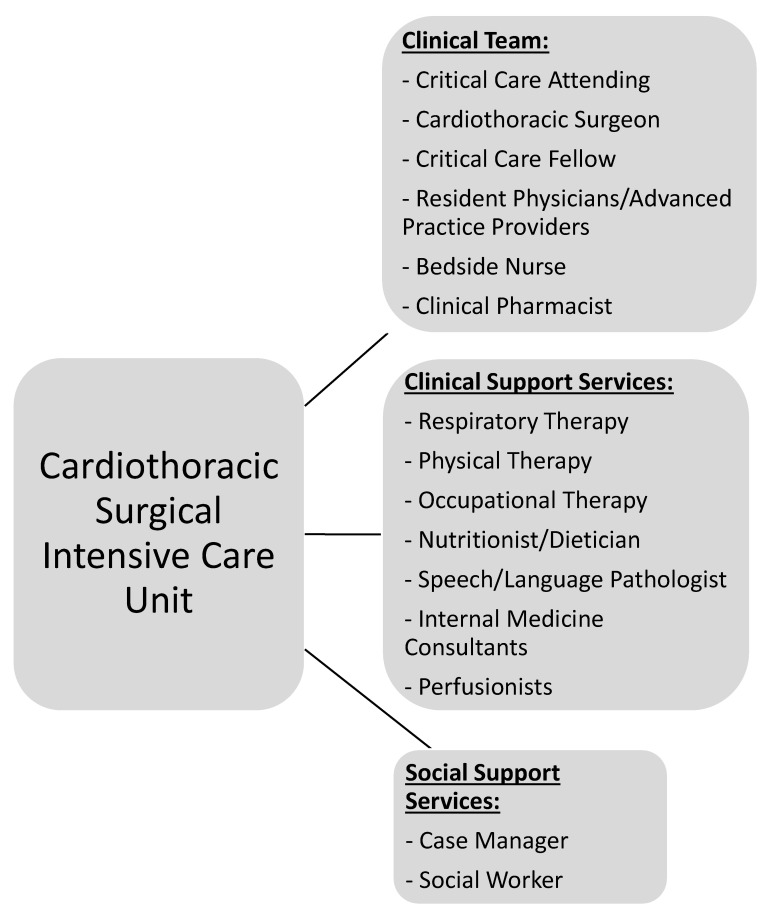
Representation of the most common network of team members required to be involved in patient care and multidisciplinary rounds in the CT-ICU due to the complex nature of critical illness.

**Figure 2 medicina-59-00047-f002:**
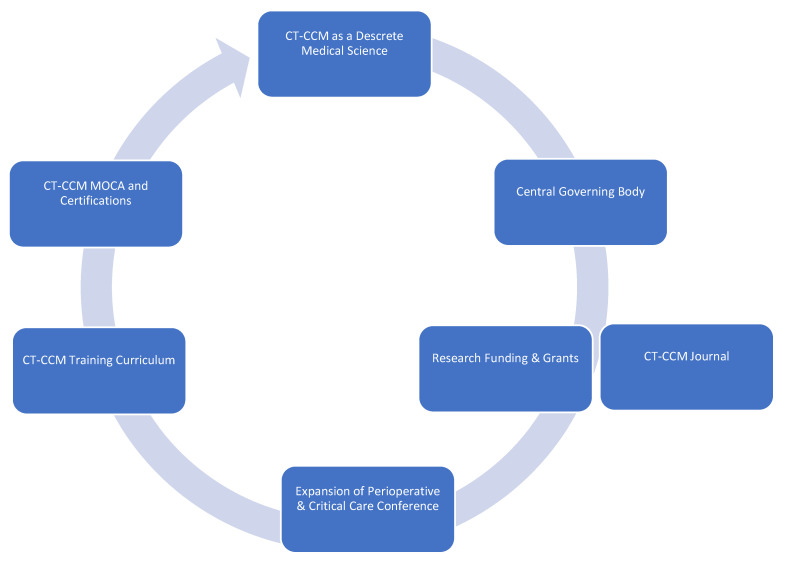
Framework of major tenets and actionable items of Cardiothoracic Surgical Critical Care Medicine.

**Figure 3 medicina-59-00047-f003:**
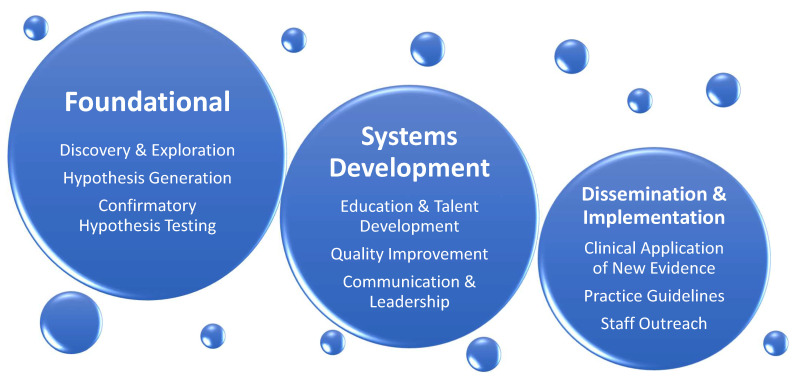
General Categories of Research Needed to Advance Cardiothoracic Surgical Critical Care Medicine. The expansion of knowledge in the field will depend on strong foundational research concentrating on basic science, resulting in new discoveries, hypothesis generation, and confirmatory testing. Concomitantly, inquiry and investigation of educational, quality improvement, and communication and leadership initiatives must occur. Finally, foundational and systems development studies need to be followed by dissemination and implementation research, where knowledge obtained from other two categories can be translated into clinical practice.

**Table 1 medicina-59-00047-t001:** Barriers to Knowledge Expansion in Cardiothoracic Surgical Critical Care Medicine.

Historical
Failure of unification of critical care medicine in the 1980s, creating specialty silos
Operating room economic incentives superior to critical care, limiting interest in subspecializing
Balanced Budget Act of 1997 caping residency spots, reducing pool of candidates available to pursue critical care
Underappreciation of importance of postoperative care on overall outcomes
Absence of recognition of CT-CCM as a unique medical science
Surgical dominance of the field
Scientific
Shortage of scientists and mentors specializing in CT-CCM specific research
Deficiency in well-established animal models specific to CT-ICU patient populations
Scarcity in hypothesis-generating research specific to CT-CCM
Paucity of CT-CCM translational research
Lack of dissemination and implementation research
Systemic/Organizational
Absence of a central governing body responsible for promotion and cultivation of CT-CCM
Knowledge silos resulting from wide array of subspecialties and societies involved in CT-CCM
Clinical and administrative workload limiting individual’s bandwidth for research projects
Educational
Absence of well-defined, unified CT-CCM training curriculum
Shortage of mentorship promoting CT-CCM inquiry
Deficiency in quality improvement training
Financial
Prohibitive costs of creating cardiopulmonary bypass animal models
Insufficient funding of CT-CCM specific research

**Table 2 medicina-59-00047-t002:** List and Summary of Publications Important to Cardiothoracic Surgical Critical Care Medicine.

Title	Authors	Year	Journal	Findings
Cardiothoracic Surgical Critical Care Leadership and Training				
Pro: Cardiothoracic Anesthesiologists Should Run Postcardiac Surgical Intensive Care Units [38]	Weiss, S.J.	2004	*JCVA*	Pro and con debate about cardiothoracic anesthesiologists running CT-ICUs
Con: Cardiothoracic Anesthesiologists Should Not Run Postcardiac Surgical Intensive Care Units [39]	Ramsey, J.	2004	*JCVA*	Pro and con debate about cardiothoracic anesthesiologists running CT-ICUs
The Emerging Specialty of Cardiothoracic Surgical Critical Care: The Leadership Role of Cardiothoracic Surgeons on the Multidisciplinary Team [16]	Katz, N.M.	2007	*JTCVS*	Editorial on CT-CCM as a new specialty and importance of CT surgeons in CT-ICU leadership
The Evolution of Cardiothoracic Critical Care [40]	Katz, N.M.	2011	*JTCVS*	Editorial on importance of CT-CCM and leading role of a CT surgeon
The Thoracic Surgical Intensivist: The Best Critical Care Doctor for Our Thoracic Surgical Patients [41]	Whitson, B.A. and D’Cunha, J.	2011	*Semin. Thorac. Cardiovasc. Surg.*	Editorial on recognition of critical care as integral component of cardiac surgery with surgeons as leaders
Cardiothoracic Surgical Critical Care: Principles, Goals and Direction [42]	Sherif, H.M.	2012	*Int. J. Surg.*	Editorial on CT-CCM as distinct discipline, its basic principles, and future directions
Developing A Curriculum for Cardiothoracic Surgical Critical Care: Impetus and Goals [43]	Sherif, H.M.	2012	*JTCVS*	Sample curriculum for surgical CT-CCM training
It Is Time for Certification In Cardiothoracic Critical Care [44]	Katz, N.M.	2013	*JTCVS*	Editorial calling for unique cardiothoracic surgical certification in critical care
The American Board of Thoracic Surgery: Update [45]	Calhoon, J.H.	2013	*JTCVS*	Official ABTS statement regarding all the certifications provided by the board. Additionally, addresses critical care pathways for surgeons and decline development of ABTS CCM certification.
Critical Care: American Board of Thoracic Surgery Update [46]	Baumgartner, W.A. et al.	2013	*JTCVS*	ABTS explaining its reasoning why it will not support certification in cardiothoracic critical care, written in response to Katz, 2013.
Certification in Cardiothoracic Surgical Critical Care [47]	Sherif, H.M., and L.H. Cohn	2014	*JTCVS*	Editorial in response to Katz 2013 supporting development of certification by ABTS
Meeting The Expanded Challenges of The Cardiothoracic Intensive Care Unit [48]	Katz, N.M.	2015	*JTCVS*	Editorial addressing changes in organization and technology in CT-ICUs, with surgical leadership at the forefront.
Is Cardiac Anaesthesiologist The Best Person to Look After Cardiac Critical Care? [49]	Mehta, Y.	2015	*Ann. Card. Anaesth.*	Editorial outlining benefits of cardiac anesthesiologists as CT-ICU intensivists
Cardiothoracic Surgical Critical Care Certification: A Future Of Distinction [50]	Sherif, H.M.	2016	*JTCVS*	Editorial highlighting the need for CT-CCM certification within cardiothoracic surgery board
Cardiothoracic Surgical Critical Care Surgeons: Many Of The Few [51]	Sherif, H.M.	2016	*JTCVS*	Letter to the editor in repones to N.D. Andersen, highlighting benefits of establishing CT-CCM as a subspecialty
Certification in Cardiothoracic Surgical Critical Care: A Distinction For Some Or For All? [52]	Andersen, N.D.	2016	*JTCVS*	Call for CT-CCM surgical certification process attainable by current and future surgeons
Cardiothoracic Surgical Critical Care Is Critical to Cardiothoracic Surgery [53]	Whitson, B.A.	2016	*JTCVS*	Letter to the editor highlighting importance of critical care to practice of cardiothoracic surgery
Redifining Our Cardiothoracic Surgical Intensive Care Units: Change is Good [54]	Chan, E.G., and J. D’Cunha	2016	*JTCVS*	Letter to the editor from ABTS members outline steps needed to advance the process of CT-CCM certification
Cardiothoracic Critical Care: A New Specialty [55]	Andrews, M.C. et al.	2017	*ASA Monitor*	Editorial highlighting benefits of dual training in cardiothoracic and critical care anesthesiology
Cardiothoracic Anesthesia and Critical Care: An Ever-Changing (and Evolving) Field [56]	Bartels, K., and S.J. Dieleman	2019	*Anes. Clin.*	Preface to Special Issue of the journal centered on cardiothoracic anesthesia and critical care
Evolving role of anesthesiology intensivists in cardiothoracic critical care [57]	Shelton, K.T. and J.P. Wiener-Kronish,	2020	*Anesthesiology*	Editorial highlighting cardiothoracic surgical intensivists at Massachusetts General Hospital
Staffing of CT-ICUs				
Cardiothoracic Intensive Care: Operation and Administration [58]	Savino, J.S. et al.	2000	*Semin. Thorac. Cardiovasc. Surg.*	Review article outlining emerging importance of physicians dedicated to postoperative medical and surgical management.
Quality Improvement Program Decreases Mortality After Cardiac Surgery [59]	Stamou, S.C. et al.	2008	*JTCVS*	Single center retrospective analysis of outcomes before and after implementation of quality improvement program, including multidisciplinary rounding involving intensivists. Implementation was associated with a decrease in mortality.
Continous Quality Improvement Program and Major Morbidity After Cardiac Surgery [60]	Stamou, S.C. et al.	2008	*Am. J. Cardiol.*	Single-center retrospective analysis of continuous quality improvement program including multidisciplinary involvement and intensivists rounding decreased sepsis and cardiac tamponade
Quality Improvement Program Increases Early Tracheal Extubation Rate and Decreases Pulmonary Complications and Resource Utilization After Cardiac Surgery	Camp S.L. et al.	2009	*J. Card. Surg.*	Single center retrospective analysis of quality improvement program implementation increased early extubation and decreased pulmonary complications
Impact of 24-Hour In-House Intensivists on a Dedicated Cardiac Surgery Intensive Care Unit [61]	Kumar, K. et al.	2009	*Ann. Thorac. Surg.*	Retrospective cohort study of 24 h in-house intensivist coverage associated with reduced hospital stay, transfusions, and requirement for mechanical ventilation
Cardiothoracic Surgeon Management of Postoperative Cardiac Critical Care [62]	Withman, G.J. et al.	2011	*JAMA*	Retrospective data review of patients after cardiac surgery where noncardiac intensivists were changed to cardiothoracic surgeons showing decreased length of stay and decrease drug cost
The Benefits of 24/7 In-House Intensivist Coverage For Prolonged-Stay Cardiac Surgery Patients [63]	Kumar, K.	2014	*JTCVS*	Retrospective before-and-after observational study assessing outcomes in patients requiring prolonged CT-ICU stay after implementation of 24/7 in-house intensivists. Reduction in transfusions, ICU complications, total hospital stay, but no changes in ICU stay or 30-day mortality were observed.
Postoperative Complications and Outcomes Associated with a Transition to 24/7 Intensivist Management of Cardiac Surgery Patients [64]	Benoit, M.A. et al.	2017	*Crit. Care Med.*	Retrospective before-and-after observational study comparing outcomes between night resident coverage to 24/7 in-house intensivists coverage. Change was associated with reduction in major postoperative complications, duration of mechanical ventilation, CT-ICU readmissions, and surgical postponement.
Does The Full-Time Presence of An Intensivist Lead to Better Outcomes in The Cardiac Surgical Intensive Care Unit? [65]	Huard, P. et al.	2020	*JTCVS*	Retrospective before-and-after study comparing outcomes nighttime resident/fellow coverage to 24 h intensivist coverage. Implementation reduced mortality in patients with expected operative mortality of ≥5%, duration of mechanical ventilation, and the risk of prolonged ventilation.
Influence of High-Intensity Staffing Model in a Cardiac Srugery Intensive Care Unit on Postoperative Clinical Outcomes [66]	Lim, J.Y. et al.	2020	*JTCVS*	Retrospective before-and-after analysis comparing resident ran service to daytime intensivists and night resident. Implementation reduced readmissions, infections, transfusions, but did not affect 30-day mortality.
Survey of Contemporary Cardiac Surgery Intensive Care Unit Models in The United States [67]	Arora, R.C. et al.	2020	*Ann. Thorac. Surg.*	Survey of current staffing models in CT-ICUs in the US. 47% open, 41% semi-open, and 12% closed. 67% were pulmonary/CCM and 44% of after-hours providers were physician assistants.
The Presence of A Dedicated Cardiac Surgical Intensive Care Service Impacts Clinical Outcomes in Adult Cardiac Surgery Patients [68]	Lee, L.S. et al.	2022	*J. Card. Surg.*	Retrospective before-and-after study assessing outcomes after implementation of intensive care service. Length of stay, duration of mechanical ventilation, and renal failure were significantly reduced, with greatest improvement in CABG patients.
Importance of High-Performing Teams in the Cardiovascular Intensive Care Unit [69]	Kennedy-Metz, L.R. et al.	2022	*JTCVS*	Expert editorial on high-functioning clinical teams relating to CT-ICU practice.
Nationwide Clinical Practice Patterns of Anesthesiology Critical Care Physicians—A Survey to Members of The Society of Critical Care Anesthesiologists [70]	Shaefi, S. et al.	2022	*Anesth. & Analg.*	Nationwide survey of critical care anesthesiologist showing that nearly 70% practice in CT-ICUs.
Selected CT-ICU Knowledge Reviews				
Critical Care of the Cardiac Patient [71]	Tung, A.	2013	*Anesthesiol. Clin.*	Review of rapidly evolving areas of CT-ICU care: mechanical ventilation, transfusion thresholds, hemodynamic monitoring, and central line insertion
Cardiothoracic Critical Care [72]	Lobdell, K.W. et al.	2017	*Surg. Clin. N. Am.*	Review of CT-CCM concentrated on high-performing teams, system, and culture, demanding proactive, interactive, precise, and expert team.
Advances in Critical Care Management of Patients Undergoing Cardiac Surgery [73]	Aneman, A. et al.	2018	*Intensive Care Med.*	Narrative review outlining standards of care, recent advances, and future areas of research in the critical care management of cardiothoracic patients.
End-of-Life Care in Cardiothoracic Surgery [74]	Birriel, B., and K. D’Angelo	2019	*Crit. Care Nurs. Clin. N. Am.*	Review of literature on end-of-line care in CT-ICU
Dissemination and Implementation Science in Cardiothoracic Surgery: A Review and Case Study [75]	Heiden, B.T. et al.	2022	*Ann. Thorac. Surg.*	Expert review of dissemination and implementation science in the context of cardiothoracic surgeon, providing tools to implement evidence based practice.

ABTS, American Board of Thoracic Surgery; CABG; coronary artery bypass grafting; CCM, critical care medicine; CT-ICU, cardiothoracic intensive care; CT-CCM, cardiothoracic surgical critical care medicine; ICU, intensive care unit.

## Data Availability

No data sets, or research data. Not applicable.

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
