# Peer review of "The Future of Cardiothoracic Surgical Critical Care Medicine as a Medical Science: A Call to Action"

_medicina, 2022, doi:10.3390/medicina59010047_

Round 1

Reviewer 1 Report

The reviewer agrees with most context of the manuscript. Indeed, a review of CT-CCM is in requirement and helpful for surgeon, cardiologist, or nursing staff. However, the review ignored pediatric population who needs the care of CT-CCM. First, with the advancement of prenatal diagnostic techniques and pediatric cardiac surgical techniques, the number of children born with congenital heart disease is gradually decreasing but the number of such children growing into adult is increasing. Second, How to take care such patients? Whether the inherent or postoperative hemodynamic abnormalities of children with congenital heart disease, such as pressure overload or volume overload, affect the maturation of their hearts, and ultimately their life quality?

The reviewer hope that the authors will added pediatric CT-CCM to make the review more complete. 

Author Response

We thank the Reviewer for their interest in our manuscript and appreciate their specific revisions outlined below

  • The reviewer agrees with most context of the manuscript. Indeed, a review of CT-CCM is in requirement and helpful for surgeon, cardiologist, or nursing staff. However, the review ignored pediatric population who needs the care of CT-CCM. First, with the advancement of prenatal diagnostic techniques and pediatric cardiac surgical techniques, the number of children born with congenital heart disease is gradually decreasing but the number of such children growing into adult is increasing. Second, How to take care such patients? Whether the inherent or postoperative hemodynamic abnormalities of children with congenital heart disease, such as pressure overload or volume overload, affect the maturation of their hearts, and ultimately their life quality?
  • The reviewer hope that the authors will added pediatric CT-CCM to make the review more complete. 
    • We thank the reviewer for their insightful comments. As the primary focus of the special issue, as well as the CT-CCM review focuses on the adult population and the unique challenges and requirements necessary to manage that population; pediatrics was not addressed.  Pediatrics poses an array of alternative considerations as well as treatment modalities which remain outside of the scope of this review and our expertise, as we are adult intensivists.
  • We added word “adult” in line 54 to ensure readers are not confused.

Reviewer 2 Report

This is a review paper regarding the current state and future goals of cardiothoracic surgical intensive care medicine. The review addresses knowledge gaps and directions to future scientific endeavors.

Comments

1. General Comment: Tables and Figures are necessarily needed. E.g., a table with all published guidelines regarding cardiothoracic surgical intensive care medicine or a table with the most important clinical trials. Or a figure with an algorithm or diagram of main knowledge gaps or a network of who should work in the cardiothoracic surgical intensive care unit. The presence of such illustrations (not only one… consider adding 2-3 tables/figures) is imperative.

2. You could also add a third part with some direct 2–3-line opinions/perceptions/interviews of medical personnel working in CT-ICUs

3. Since interventional cardiology emerged, a lot of medical conditions were treated with interventional and non-surgical procedures. How did this trend affect the work of cardiothoracic surgical intensive care units? Eg less CABGs? Or change in age distributions? Since when?

4. Page 98 abbreviation CT-CCM?

5. Line 191-192: “Acute kidney injury is the most common major complication occurring after cardiac surgery with incidence reaching 40%” – shouldn’t this be the first condition to be reported (before delirium) since it is the most common?

6. Line 304-310: What is finally the strategy recommended regarding the hemoglobin threshold for transfusion?

7. Line 347 “Paralysis After Aortic Aneurysm Surgery” – what is the role of physiotherapy?

8. What was the impact of COVID-19 on CT-ICUs?

9. Line 674 “patients or their families have the right to refuse therapies at any point” – that’s the ethical part that we should respect our patients’ wishes. What about the legal part? Do patients sign a form if they do not wish any further procedures or medical handling? What if the patient cannot sign etc?

Author Response

Reviewer 1: We thank the Reviewer for their interest in our manuscript and appreciate their specific revisions outlined below. We have corrected the formatting of our references to meet the journal’s guidelines.

  • Since interventional cardiology emerged, a lot of medical conditions were treated with interventional and non-surgical procedures. How did this trend affect the work of cardiothoracic surgical intensive care units? Eg less CABGs? Or change in age distributions? Since when?

    • Since the advancement of percutaneous coronary and structural heart interventions there have been variable outcomes.  There remains a certain subset of patients who are either not amenable or have unsuccessful percutaneous therapies. Some of these patients can require overt full mechanical circulatory support [ECLS].  These patients with complex valvular and cardiac disease are best-served in a CT-ICU managed by cardiothoracic intensivists familiar with their physiology.  In addition, as the aging population being treated continues to increase, so too does the severity and chronicity of disease.  Even the treatment of coronary artery disease is still heavily relying on a surgical intervention (CABG), especially in patients with comorbidities such as diabetes, or with complex lesions.1 This continuing necessity of the cardiothoracic intensivist in managing these complex patients is unlikely to go away any time soon, but in fact, increase over the next decades. This will be driven by:
      • Aging baby-boomer population with increased burden of disease
        • This will result in higher degree of coronary artery disease and ischemic cardiomyopathy patients requiring care.
          • A subset of these patients will require durable mechanical support, such as LVADs, or heart transplants.
            • Recent developments in xenotransplantation2 gives hope of an increase in organ availability. These patients will require significant CT-ICU care.
            • Other will only qualify for destination therapy with LVADs.3
      • Increase in lung failure and lung transplantation need
      • Increasing number of patients with congenital heart lesions who survived until adulthood, who will require redo surgeries, or heart transplantations
      • Unforeseen events like a pandemic
    • We have added a section in our introduction addressing these points. Lines 38-42 address this comment and are highlighted in in Tracked Version of the manuscript.
  1. Welt FGP. CABG versus PCI - End of the Debate?. N Engl J Med. 2022;386(2):185-187. doi:10.1056/NEJMe2117325
  2. Griffith BP, Goerlich CE, Singh AK, et al. Genetically Modified Porcine-to-Human Cardiac Xenotransplantation. N Engl J Med. 2022;387(1):35-44. doi:10.1056/NEJMoa2201422
  3. Michaels A, Cowger J. Patient Selection for Destination LVAD Therapy: Predicting Success in the Short and Long Term. Curr Heart Fail Rep. 2019;16(5):140-149. doi:10.1007/s11897-019-00434-1
  4. Khakban, A., et al., The projected epidemic of chronic obstructive pulmonary disease hospitalizations over the next 15 years. A population-based perspective. American journal of respiratory and critical care medicine, 2017. 195(3): p. 287-291.
  5. Sauleda, J., et al., Idiopathic pulmonary fibrosis: epidemiology, natural history, phenotypes. Medical Sciences, 2018. 6(4): p. 110.
  6. Nathan, S.D., The future of lung transplantation. Chest, 2015. 147(2): p. 309-316.
  7. Azzi, M., et al., Extracorporeal CO2 removal in acute exacerbation of COPD unresponsive to non-invasive ventilation. BMJ Open Respir Res, 2021. 8(1).
  • What was the impact of COVID-19 on CT-ICUs?
    • Thank you for this comment. The COVID-19 pandemic placed strains on modern-day critical care medicine unlike any other period in history. The CT-ICU in addition to all the other critical care areas served as epicenters for advanced supportive care.  Many centers which utilized extracorporeal life support in severe cases were staffed solely by cardiothoracic intensivists with little to no other medical provider assistance; requiring their knowledge base and care provided to be at its highest levels.  Additionally, the COVID-19 pandemic forced cardiothoracic intensivists to analyze global data for scarce resource allocation as well as command end-of-life care conferences in the most severe of circumstances; all while dealing with a relatively unknown disease process.
  • Page 98 abbreviation CT-CCM?
    • Thank you for this comment. CT-CCM has been defined in the first sentence of the manuscript.
  • Line 191-192: “Acute kidney injury is the most common major complication occurring after cardiac surgery with incidence reaching 40%” – shouldn’t this be the first condition to be reported (before delirium) since it is the most common?
    • Thank you for this comment. We changed the sequence per your suggestion. Please see Tracked Version reflecting this change in red.
  • Line 304-310: What is finally the strategy recommended regarding the hemoglobin threshold for transfusion?
    • Thank you for this comment. The best studies we have state that threshold <7.5 is safe, with threshold of 7.0 not explicitly studies. We have reflected this in Tracked Version, with new comments added in red.
    • 437-447: For the general cardiac surgical population, a threshold of <7.5 g/dL is safe: a 5,243-participant study of a <7.5 g/dL versus a <9.5 g/dL trigger for transfusion during and after moderate-to-high-risk cardiothoracic surgery found reduced transfusions and no evidence of harm in the restrictive group, with consistent outcomes at six monthsIt is quite likely that a threshold of 7.0 is comparable to 7.5, but that has not been explicitly studied in large cardiac surgery trials. For non-surgical patients with active ischemia, a threshold between 7-8 g/dL is likely safe, but not firmly established: a trial of 668 patients with anemia and myocardial infarction found that a transfusion trigger of <7 g/dL was noninferior to a trigger of 10 g/dL for major adverse cardiac events at 30 days – although the confidence interval for this result may include clinically significant harm The results of the 3,500-patient Myocardial Ischemia and Transfusion (MINT) trial, expected to
  • “patients or their families have the right to refuse therapies at any point” – that’s the ethical part that we should respect our patients’ wishes. What about the legal part? Do patients sign a form if they do not wish any further procedures or medical handling? What if the patient cannot sign etc?
    • Thank you for your comment. The legality of the situation is the same as the ethical suggestions. For example, formal change in code status does not require a signature rather just a conversation with the care team. Patient and their surrogate decision makers have the right to refuse any treatment at any time, even if it is recommended by the medical team. In fact, without signed consent the patient would not undergo any treatment, particularly invasive procedures.
  • Line 347 “Paralysis After Aortic Aneurysm Surgery” – what is the role of physiotherapy?
    • Thank you for this comment. Physical therapists play an integral role in the recovery of our patients with spinal cord ischemia after aortic surgery. The interventions utilized by these physical therapists are aimed at guiding locomotion utilizing exercise therapy and other similar measures. There is a lack of randomized controlled trials, however, describing the efficacy of any single physical therapy intervention. This is yet another gap in the knowledge of treating patients with this condition. Future studies are required which investigate different types of physical therapy interventions at improving the recovery after aortic surgery, and if there is a role of physical therapy in reversing the pathophysiology and return of functional status in these patients.
    •  
  • You could also add a third part with some direct 2–3-line opinions/perceptions/interviews of medical personnel working in CT-ICUs
    • Thank you for this comment. Given the broad topic of scientific knowledge gaps and future of CT-CCM as its own science, we find that adding Part 3 with opinions/perceptions/interviews inappropriate for the topic. Please see our other manuscript included in this special issue of Medicina titled “Developing of Cardiothoracic Surgical Critical Care Intensivists: A Case for Distinct Training” addressing this topic.
  • General Comment: Tables and Figures are necessarily needed. E.g., a table with all published guidelines regarding cardiothoracic surgical intensive care medicine or a table with the most important clinical trials. Or a figure with an algorithm or diagram of main knowledge gaps or a network of who should work in the cardiothoracic surgical intensive care unit. The presence of such illustrations (not only one… consider adding 2-3 tables/figures) is imperative.
    • Thank for your insightful comments. Please see new tables and figures included in the manuscript. Copies are provided below.

Table 1. Barriers to Knowledge Expansion in Cardiothoracic Surgical Critical Care Medicine 

                                                                       Legacy         

Failure of unification of critical care medicine in 1980’s, creating specialty silos 

Underappreciation of importance of postoperative care on overall outcomes

Absence of recognition of CT-CCM as a unique medical science

Surgical dominance of the field

Scientific

Deficiency in scientists and mentors specializing in CT-CCM specific research

Deficiency in well-established animal models specific to CT-ICU patient populations

Deficiency in hypothesis-generating research specific to CT-CCM

Deficiency in CT-CCM translational research 

Deficiency in dissemination and implementation infrastructure

Systemic/Organizational

Absence of a central governing body responsible for promotion and cultivation of CT-CCM

Knowledge silos resulting from wide array of subspecialties and societies involved in CT-CCM

Clinical and administrative workload limiting individual’s bandwidth for research projects

Educational

Absence of well-defined, unified CT-CCM training curriculum

Deficiency in mentorship promoting CT-CCM inquiry

Deficiency in quality improvement training

Financial

Prohibitive costs of creating cardiopulmonary bypass animal models

Deficiency in CT-CCM specific funding

Legend. CT-CCM, Cardiothoracic Surgical Critical Care Medicine

Figure 1: Representation of the most common network of team members required to be involved in patient care and multidisciplinary rounds in the CTICU due to the complex nature of critical illness.

Figure 2. Framework of major tenants and actionable items of CT-CCM

Figure 3. General Categories of Research Gaps in the Field of CT-CCM

Table 3. Summary of Specific Knowledge Gaps and Research Goals in CT-CCM

Title

Authors

Year

Journal

Findings

Cardiothoracic Surgical Critical Care Leadership and Training

Pro: Cardiothoracic Anesthesiologists Should Run Postcardiac Surgical Intensive Care Units

Weiss, S.J.

2004

JCVA

Pro and con debate about cardiothoracic anesthesiologists running CTICUs

Con: Cardiothoracic Anesthesiologists Should Not Run Postcardiac Surgical Intensive Care Units

Ramsey, J.

2004

JCVA

Pro and con debate about cardiothoracic anesthesiologists running CTICUs

The Emerging Specialty of Cardiothoracic Surgical Critical Care: The Leadership Role of Cardiothoracic Surgeons on the Multidisciplinary Team

Katz, N.M.

2007

JTCVS

Editorial on CT-CCM as a new specialty and importance of CT surgeons in CTICU leadership

The Evolution of Cardiothoracic Critical Care

Katz, N.M.

2011

JTCVS

Editorial on importance of CTCCM and leading role of a CT surgeon

The Thoracic Surgical Intensivist: The Best Critical Care Doctor for Our Thoracic Surgical Patients

Whitson, B.A. and D’Cunha, J.

2011

Semin Thorac Cardiovasc Surg

Editorial on recognition of critical care as integral component of cardiac surgery with surgeons as leaders

Cardiothoracic Surgical Critical Care: Principles, Goals and Direction

Sherif, H.M.

2012

Int J Surg

Editorial on CTCCM as distinct discipline, its basic principles, and future directions

Developing A Curriculum for Cardiothoracic Surgical Critical Care: Impetus and Goals

Sherif, H.M.

2012

JTCVS

Sample curriculum for surgical CTCCM training

It Is Time for Certification In Cardiothoracic Critical Care

Katz, N.M.

2013

JTCVS

Editorial calling for unique cardiothoracic surgical certification in critical care

The American Board of Thoracic Surgery: Update

Calhoon, J.H.

2013

JTCVS

Official ABTS statement regarding all the certifications provided by the board. Also addresses critical care pathways for surgeons and decline development of ABTS CCM certification.

Critical Care: American Board of Thoracic Surgery Update

Baumgartner, W.A., et al

2013

JTCVS

ABTS explaining its reasoning why it will not support certification in cardiothoracic critical care, written in response to Katz, 2013.

Certification in Cardiothoracic Surgical Critical Care

Sherif, H.M., and L.H. Cohn

2014

JTCVS

Editorial in response to Katz 2013 supporting development of certification by ABTS 

Meeting The Expanded Challenges of The Cardiothoracic Intensive Care Unit

Katz, N.M.

2015

JTCVS

Editorial addressing changes in organization and technology in CTICUs, with surgical leadership at the forefront.

Is Cardiac Anaesthesiologist The Best Person to Look After Cardiac Critical Care?

Mehta, Y.

2015

Ann Card Anaesth

Editorial outlining benefits of cardiac anesthesiologists as CT-ICU intensivists

Cardiothoracic Surgical Critical Care Certification: A Future Of Distinction

Sherif, H.M.

2016

JTCVS

Editorial highlighting the need for CTCCM certification within cardiothoracic surgery board

Cardiothoracic Surgical Critical Care Surgeons: Many Of The Few

Sherif, H.M.

2016

JTCVS

Letter to the editor in repones to N.D. Andersen, highlighting benefits of establishing CTCCM as a subspecialty

Certification in Cardiothoracic Surgical Critical Care: A Distinction For Some Or For All?

Andersen, N.D.

2016

JTCVS

Call for CTCCM surgical certification process attainable by current and future surgeons

Cardiothoracic Surgical Critical Care Is Critical to Cardiothoracic Surgery

Whitson, B.A.

2016

JTCVS

Letter to the editor highlighting importance of critical care to practice of cardiothoracic surgery

Redifining Our Cardiothoracic Surgical Intensive Care Units: Change is Good

Chan, E.G., and J. D’Cunha

2016

JTCVS

Letter to the editor from ABTS members outline steps needed to advance the process of CTCCM certification 

Cardiothoracic Critical Care: A New Specialty

Andrews, M.C., et al

2017

ASA Monitor

Editorial highlighting benefits of dual training in cardiothoracic and critical care anesthesiology

Cardiothoracic Anesthesia and Critical Care: An Ever-Changing (and Evolving) Field

Bartels, K., and S.J. Dieleman

2019

Anes Clin

Preface to Special Issue of the journal centered on cardiothoracic anesthesia and critical care

Evolving role of anesthesiology intensivists in cardiothoracic critical care

Shelton, K.T. and J.P. Wiener-Kronish,

2020

Anesthesiology

Editorial highlighting cardiothoracic surgical intensivists at Massachusetts General Hospital

Staffing of CT-ICUs

Cardiothoracic Intensive Care: Operation and Administration

Savino, J.S., et al

2000

Semin Thorac Cardiovasc Surg

Review article outlining emerging importance of physicians dedicated to postoperative medical and surgical management.

Quality Improvement Program Decreases Mortality After Cardiac Surgery

Stamou, S.C., et al

2008

JTCVS

Single center retrospective analysis of outcomes before and after implementation of quality improvement program, including multidisciplinary rounding involving intensivists. Implementation was associated with a decrease in mortality.

Continous Quality Improvement Program and Major Morbidity After Cardiac Surgery

Stamou, S.C., et al

2008

Am J Cardiol

Single center retrospective analysis of continuous quality improvement program including multidisciplinary involvement and intensivists rounding decreased sepsis and cardiac tamponade

Quality Improvement Program Increases Early Tracheal Extubation Rate and Decreases Pulmonary Complications and Resource Utilization After Cardiac Surgery

Camp S.L., et al

2009

J Card Surg

Single center retrospective analysis of quality improvement program implementation increased early extubation and decreased pulmonary complications

Impact of 24-Hour In-House Intensivists on a Dedicated Cardiac Surgery Intensive Care Unit

Kumar, K., et al

2009

Ann Thorac Surg

Retrospective cohort study of 24-hour in-house intensivist coverage associated with reduced hospital stay, transfusions, and requirement for mechanical ventilation  

Cardiothoracic Surgeon Management of Postoperative Cardiac Critical Care

Withman, G.J., et al

2011

JAMA

Retrospective data review of patients after cardiac surgery where noncardiac intensivists were changed to cardiothoracic surgeons showing decreased length of stay and decrease drug cost

The Benefits of 24/7 In-House Intensivist Coverage For Prolonged-Stay Cardiac Surgery Patients

Kumar, K.

2014

JTCVS

Retrospective before-and-after observational study assessing outcomes in patients requiring prolonged CT-ICU stay after implementation of 24/7 in-house intensivists. Reduction in transfusions, ICU complications, total hospital stay, but no changes in ICU stay or 30-day mortality were observed.

Postoperative Complications and Outcomes Associated with a Transition to 24/7 Intensivist Management of Cardiac Surgery Patients

Benoit, M.A., et al

2017

Crit Care Med

Retrospective before-and-after observational study comparing outcomes between night resident coverage to 24/7 in-house intensivists coverage. Change was associated with reduction in major postoperative complications, duration of mechanical ventilation, CT-ICU readmissions, and surgical postponement.

Does The Full-Time Presence of An Intensivist Lead to Better Outcomes in The Cardiac Surgical Intensive Care Unit?

Huard, P., et al

2020

JTCVS

Retrospective before-and-after study comparing outcomes nighttime resident/fellow coverage to 24-hour intensivist coverage. Implementation reduced mortality in patients with expected operative mortality of ≥5%, duration of mechanical ventilation, and the risk of prolonged ventilation.

Influence of High-Intensity Staffing Model in a Cardiac Srugery Intensive Care Unit on Postoperative Clinical Outcomes

Lim, J.Y., et al

2020

JTCVS

Retrospective before-and-after analysis comparing resident ran service to daytime intensivists and night resident. Implementation reduced readmissions, infections, transfusions, but did not affect 30-day mortality.

The Presence of A Dedicated Cardiac Surgical Intensive Care Service Impacts Clinical Outcomes in Adult Cardiac Surgery Patients

Lee, L.S., et al

2022

J Card Surg

Retrospective before-and-after study assessing outcomes after implementation of intensive care service. Length of stay, duration of mechanical ventilation, and renal failure were significantly reduced, with greatest improvement in CABG patients.

Survey of Contemporary Cardiac Surgery Intensive Care Unit Models in The United States

Arora, R.C., et al

2022

Ann Thorac Surg

Survey of current staffing models in CT-ICUs in the US. 47% open, 41% semi-open, and 12% closed. 67% were pulmonary/CCM and 44% of after-hours providers were physician assistants.

Importance of High-Performing Teams in the Cardiovascular Intensive Care Unit

Kennedy-Metz, L.R., et al

2022

JTCVS

Expert editorial on high-functioning clinical teams relating to CT-ICU practice.

Nationwide Clinical Practice Patterns of Anesthesiology Critical Care Physicians—A Survey to Members of The Society of Critical Care Anesthesiologists

Shaefi, S., et al

2022

Anesth & Analg

Nationwide survey of critical care anesthesiologist showing that nearly 70% practice in CT-ICUs.

Selected CT-ICU Knowledge Reviews

Critical Care of the Cardiac Patient

Tung, A.

2013

Anesthesiol Clin

Review of rapidly evolving areas of CT-ICU care: mechanical ventilation, transfusion thresholds, hemodynamic monitoring, and central line insertion

Cardiothoracic Critical Care

Lobdell, K.W., et al

2017

Surg Clin North Am

Review of CT-CCM concentrated on high-performing teams, system, and culture, demanding proactive, interactive, precise, and expert team.

Advances in Critical Care Management of Patients Undergoing Cardiac Surgery

Aneman, A., et al

2018

Intensive Care Med

Narrative review outlining standards of care, recent advances, and future areas of research in the critical care management of cardiothoracic patients.

End-of-Life Care in Cardiothoracic Surgery

Birriel, B., and K. D’Angelo

2019

Crit Care Nurs Clin North Am

Review of literature on end-of-line care in CT-ICU

Dissemination and Implementation Science in Cardiothoracic Surgery: A Review and Case Study

Heiden, B.T., et al

2022

Ann Thorac Surg

Expert review of dissemination and implementation science in the context of cardiothoracic surgeon, providing tools to implement evidence based practice.

Round 2

Reviewer 2 Report

Please review the whole manuscript meticulously and perform necessary corrections - e.g., in Table 3 different font types/sizes are used, in some places CTCCM is used, in other places CT-CCM. This is very unpleasant for the reader and may convey an impression of unprofessionalism.

Author Response

We want to thank reviewer # 2 for their insightful comments. The fonts have been corrected and shortcuts have been made consistent throughout the manuscript. Please see the tracked changes in the re-submitted version of the manuscript.